# Balanced Mixed-Type Tabular Data Synthesis with Diffusion Models

**Zeyu Yang**                                                    *zeyuyang8@gmail.com*
*Department of Electrical and Computer Engineering*
*Rice University*

**Han Yu**                                                       *hy29@rice.edu*
*Department of Electrical and Computer Engineering*
*Rice University*

**Peikun Guo**                                                   *pg34@rice.com*
*Department of Electrical and Computer Engineering*
*Rice University*

**Khadija Zanna**                                                *kz35@rice.edu*
*Department of Electrical and Computer Engineering*
*Rice University*

**Xiaoxue Yang**                                                 *yolandaxyang@gmail.com*
*Department of Computing*
*Imperial College London*

**Akane Sano**                                                   *akane.sano@rice.edu*
*Department of Electrical and Computer Engineering*
*Rice University*

**Reviewed on OpenReview:** *https://openreview.net/forum?id=dvRysCqmYQ*

## Abstract

Diffusion models have emerged as a robust framework for various generative tasks, including tabular data synthesis. However, current tabular diffusion models tend to inherit bias in the training dataset and generate biased synthetic data, which may influence discriminatory actions. In this research, we introduce a novel tabular diffusion model that incorporates sensitive guidance to generate fair synthetic data with balanced joint distributions of the target label and sensitive attributes, such as sex and race. The empirical results demonstrate that our method effectively mitigates bias in training data while maintaining the quality of the generated samples. Furthermore, we provide evidence that our approach outperforms existing methods for synthesizing tabular data on fairness metrics such as demographic parity ratio and equalized odds ratio, achieving improvements of over 10%. Our implementation is available at `https://github.com/comp-well-org/fair-tab-diffusion`.

## 1 Introduction

Tabular data synthesis, particularly when involving mixed-type variables, presents unique challenges due to the need to simultaneously handle continuous and discrete features. Recent advances in diffusion models (Sohl-Dickstein et al., 2015; Ho et al., 2020) are now being leveraged to tackle these challenges. Diffusion models have demonstrated remarkable performance in generative modeling for various tasks, especially text-to-image synthesis. Researchers use diffusion frameworks to model images paired with text descriptions and show promising results (Nichol et al., 2021; Saharia et al., 2022). Advancing further from diffusion models in

pixel space, latent diffusion frameworks optimize image representation in a low-dimensional space, thereby enhancing synthetic image quality and efficiency (Rombach et al., 2022; Ramesh et al., 2022). The impressive results of diffusion models in image synthesis extended to generating sequential data such as audio and time series (Kong et al., 2020; Rasul et al., 2021; Li et al., 2022). This advancement has gradually overshadowed generative adversarial networks, which were previously a significant focus in generative modeling, as diffusion models consistently demonstrate superior performance across diverse tasks (Dhariwal & Nichol, 2021; Stypułkowski et al., 2023; Mazé & Ahmed, 2023).

Researchers have recently been motivated by the versatility of diffusion models to explore the potential in synthesizing tabular data (Kotelnikov et al., 2023; He et al., 2023). Tabular data often involves sensitive user information, which raises significant privacy concerns when used for model training. Moreover, such datasets suffer from limitations in size and imbalanced class distribution (Jesus et al., 2022), leading to challenges in training robust and fair machine learning models. To address these issues, there is a growing demand for effective tabular data synthesis. However, generative modeling of tabular data is challenging due to the combination of discrete and continuous features and their varying value distributions. Interpolation-based techniques, such as SMOTE (Chawla et al., 2002), are simple and effective but they pose privacy concerns as they generate new data points through direct interpolation of existing data. Researchers have used adversarial generative networks and variational autoencoders to generate tabular data with promising results (Ma et al., 2020; Xu et al., 2019). Research on diffusion models for tabular data is ongoing, and the primary challenge is to handle continuous and discrete features simultaneously. In Austin et al. (2021), the authors suggest using diffusion frameworks for discrete features with Markov transition matrices. Recent developments such as TabDDPM (Kotelnikov et al., 2023) and CoDi (Lee et al., 2023), which combine Markov transition functions (called diffusion kernels) for continuous and discrete features, mark significant progress in this field and underscore the potential of diffusion models as a comprehensive solution for tabular data synthesis.

Despite the success of diffusion models in various generative tasks, ethical concerns regarding potential bias and unsafe content in the synthetic data persist. Generative models are inherently data-driven, thereby propagating these biases into the synthetic data. Studies such as fair diffusion (Friedrich et al., 2023) and safe latent diffusion (Schramowski et al., 2023) have shown that latent diffusion models, particularly in text-to-image synthesis, can create images biased toward certain privileged groups and may contain inappropriate elements, such as nudity and violence. In response, they propose methods to apply hazardous or sensitive guidance in latent space and then eliminate bias and offensive elements in the synthetic images. In addition to text and images, tabular datasets frequently exhibit notable bias, often stemming from issues related to the disproportionate representation of sensitive groups. For instance, some datasets exhibit significant imbalances in demographic groups regarding sensitive attributes such as sex and race (Le Quy et al., 2022). Training machine learning models on biased tabular data can result in discriminatory outcomes for underrepresented groups. However, in our literature review, research on the safety and fairness of diffusion models in areas beyond text-to-image synthesis is limited. Although diffusion models have shown remarkable performance in tabular data synthesis (Sattarov et al., 2023; He et al., 2023; Kotelnikov et al., 2023; Lee et al., 2023; Kim et al., 2022), they are prone to learning and replicating these biases in training data.

Existing tabular diffusion models are typically either unconditional or conditioned on a single target variable. For example, CoDi (Lee et al., 2023) leverages two interconnected diffusion models to separately process continuous and discrete variables while maintaining their inter-variable correlations through co-evolutionary conditioning and contrastive learning. Although CoDi efficiently captures inter-variable relationships, its conditioning mechanism remains restricted to single-label guidance. As a result, CoDi does not explicitly address fairness considerations in the presence of sensitive attributes. In contrast, we introduce a novel tabular diffusion model that approximates mixed-type tabular data distributions, conditioned on both outcomes and multiple sensitive attributes. Our method extends tabular diffusion models from label-only conditioning to multivariate feature-level conditioning for enhancing fairness. To promote fairness, our approach generates data with balanced joint distributions of the target label and sensitive attributes. Specifically, our approach integrates sensitive guidance into the tabular diffusion model and extends it to manage multiple sensitive attributes, such as sex and race. We utilize a U-Net architecture to approximate the data distributions in the latent space during the reverse diffusion process conditioned on various variables, including the target

label and sensitive attributes. To ensure fairness while preserving the integrity of the original data, we apply the element-wise comparison technique from safe latent diffusion to regulate the sensitive guidance within an appropriate range. Additionally, we employ balanced sampling techniques to reduce disparities in group representation within the synthetic data. Given that our method leverages diffusion frameworks tailored for tabular data and supports multiple variables for conditional generation of fair tabular data, it also has the potential for extension to multi-modal data generation with fairness considerations using diffusion models. In summary, our main contributions are as follows: We employ balanced sampling techniques to reduce disparities in group representation within the synthetic data. In summary, our main contributions are as follows:

1. We introduce, to our knowledge, the first diffusion-based framework tailored to learn distributions of mixed-type tabular data with the extension from label-only conditioning to multivariate feature-level conditioning for fairness.

2. We generate tabular data that is balanced for a predefined set of sensitive attributes, addressing inherent biases in the data.

3. Our model demonstrates strong performance in terms of fidelity and diversity of synthetic data, while surpassing existing models in fairness metrics such as demographic parity ratio and equalized oddes ratio by more than 10% on average across three experimental datasets.

The remainder of this paper is organized as follows: Section 2 provides an overview of related work on fairness-aware machine learning and diffusion models. Section 3 discusses the relevant background of mixed-type modeling with diffusion models. In Section 4, we present our approach to multivariate conditioning and balanced sampling. In Section 5, we illustrate the experiments and present the results. Section 6 discusses the limitations of our method. Finally, in Section 7, we conclude our findings and discuss their implications.

## 2 Related Work

### 2.1 Bias in Machine Learning

The concept of bias in machine learning that involves assigning significance to specific features to enhance overall generalization is essential for the effective performance of models (Mehrabi et al., 2021). However, it is crucial to acknowledge that bias in machine learning can also have negative implications. Negative bias is an inaccurate assumption made by machine learning algorithms that reflects systematic or historical prejudices against certain groups of people (Zanna et al., 2022). Decisions derived from such biased algorithms can lead to adverse effects, particularly impacting specific social groups defined by factors such as sex, age, disability, race, and more.

This concept of bias and fairness in machine learning has been widely studied, the aim being to mitigate algorithmic bias of sensitive attributes from machine learning models (Mehrabi et al., 2021; Pessach & Shmueli, 2022). Many methods for bias mitigation in classification tasks have been proposed, and they can be classified into three categories: pre-processing, in-processing, and post-processing methods. Pre-processing methods (Kamiran & Calders, 2012; Calmon et al., 2017) transform the collected data to remove discrimination and train machine learning models on discrimination-free datasets. In-processing methods (Zhang et al., 2018; Agarwal et al., 2018; Kamishima et al., 2011; Zafar et al., 2017; Kamishima et al., 2012) regulate machine learning algorithms by incorporating fairness constraints or regularization terms into the objective functions. Lastly, post-processing methods (Hardt et al., 2016; Pleiss et al., 2017), implemented after training machine learning models on collected datasets, directly override the predicted labels to improve fairness.

### 2.2 Fair Data Synthesis

Fairness-aware generative models typically operate as pre-processing methods. One such method proposed by Xu et al. (2018) introduced fairness-aware adversarial generative networks that employ a fair discriminator to maintain equality in the joint probability of features and labels conditioned on subgroups within

sensitive attributes. Similarly, DECAF (Van Breugel et al., 2021), a causally-aware GAN-based framework, incorporates structural causal models into the generator to ensure fairness, enabling inference-time debiasing and compatibility with multiple fairness definitions. However, these approaches do not effectively tackle the imbalance in sensitive classes within synthetic data, and diffusion-based tabular data synthesis methods such as Kotelnikov et al. (2023); Lee et al. (2023); Zhang et al. (2023) excel in producing synthetic data of superior quality. Fair Class Balancing (Yan et al., 2020), an oversampling algorithm based on SMOTE, demonstrates impressive results but lacks flexibility in manipulating data distributions. As an interpolation-based method, it struggles with data diversity and is not well-suited for multi-modal generation. Recently, Fair4Free (Sikder et al., 2024) introduced a novel approach to fairness-aware data synthesis using data-free distillation, which trains a smaller student model to generate fair synthetic data while preserving privacy and outperforming state-of-the-art methods in fairness, utility, and data quality.

### 2.3 Fair/Safe Diffusion Models

With impressive capabilities in generative modeling, diffusion models are a class of deep generative models that utilize a Markov process that starts from noise and ends with the target data distribution. Diffusion models have demonstrated superior performance over large language models in modeling structured data like images and tables, as they are specifically designed to handle mixed-type features and maintain complex inter-dependencies (Zhang et al., 2023). Although diffusion models are widely studied, research on the fairness and safety of diffusion models is limited, and existing works (Schramowski et al., 2023; Friedrich et al., 2023) mainly focus on text-to-image synthesis tasks. Friedrich et al. (2023) utilize sensitive content, such as sex and race, within text prompts to guide training diffusion models. Subsequently, during the sampling phase, a uniform distribution of sensitive attributes is applied to generate images that are fair and do not exhibit preference towards privileged groups. Similarly, (Schramowski et al., 2023) uses unsafe text, such as nudity and violence, to guide diffusion models and remove unsafe content in the sampling phase. In addition, this method applies some techniques to remove unsafe content from synthetic images while keeping changes minimal. However, these fairness-aware diffusion models have not been adapted to tabular data synthesis, even though bias commonly exists in tabular datasets (Le Quy et al., 2022). This paper studies how to model mixed-type tabular data conditioning on labels and multiple sensitive attributes in latent space. Our method can generate balanced tabular data considering multiple sensitive attributes and subsequently achieve improved fairness scores without compromising the quality of the synthetic data.

## 3 Diffusion Models

Diffusion models, taking inspiration from thermodynamics, are probabilistic generative models that operate under the Markov assumption. Diffusion models involve two major components: a forward process and a reverse process. The forward process gradually transforms the given data $\mathbf{x}_0$ into noise $\mathbf{x}_T$ passing through $T$ steps of the diffusion kernel $q(\mathbf{x}_t|\mathbf{x}_{t-1})$. In the reverse process, $\mathbf{x}_T$ is restored to the original data $\mathbf{x}_0$ by $T$ steps of posterior estimator $p_\theta(\mathbf{x}_{t-1}|\mathbf{x}_t)$ with trainable parameters $\theta$. In this section, we explain the Gaussian diffusion kernel for continuous features and the multinomial diffusion kernel for discrete features. Furthermore, we show how to optimize the parameters $\theta$ of the posterior estimator considering mixed-type data with both continuous and discrete features.

### 3.1 Gaussian Diffusion Kernel

The Gaussian diffusion kernel is used in the forward process for continuous features. After a series of Gaussian diffusion kernels with $T$ timesteps, $\mathbf{x}_T$ follows the standard Gaussian distribution $\mathcal{N}(\mathbf{0}, \mathbf{I})$. Specifically, with the values of $\beta_t$ predefined according to a schedule such as linear, cosine, etc. (Chen, 2023), the Gaussian diffusion kernel $q(\mathbf{x}_t|\mathbf{x}_{t-1})$ can be written as:

$$q(\mathbf{x}_t|\mathbf{x}_{t-1}) = \mathcal{N}(\mathbf{x}_t|\sqrt{1-\beta_t}\mathbf{x}_{t-1}, \beta_t\mathbf{I})$$

With the starting point $\mathbf{x}_0$ known, the posterior distribution in the forward process can be calculated analytically as follows,

$$q(\mathbf{x}_{t-1}|\mathbf{x}_t, \mathbf{x}_0) = \mathcal{N}(\mathbf{x}_{t-1}|\tilde{\mu}_t(\mathbf{x}_t, \mathbf{x}_0), \tilde{\beta}_t\mathbf{I})$$

$$\tilde{\mu}_t(\mathbf{x}_t, \mathbf{x}_0) = \frac{1}{\sqrt{\alpha_t}}(\mathbf{x}_t - \frac{\beta_t}{\sqrt{1-\bar{\alpha}_t}}\epsilon) \tag{1}$$

$$\tilde{\beta}_t = \frac{1-\bar{\alpha}_{t-1}}{1-\bar{\alpha}_t}\beta_t$$

where $\alpha_t = 1 - \beta_t$, $\bar{\alpha}_t = \prod_{s=1}^{t}\alpha_s$. In the reverse process, a posterior estimator with parameters $\theta$ is used to approximate $p_\theta(\mathbf{x}_{t-1}|\mathbf{x}_t)$ at each timestep $t$ given $\mathbf{x}_t$.

$$p_\theta(\mathbf{x}_{t-1}|\mathbf{x}_t) = \mathcal{N}(\mathbf{x}_{t-1}|\hat{\mu}(\mathbf{x}_t), \hat{\boldsymbol{\Sigma}}(\mathbf{x}_t))$$

Ho et al. (2020) set the estimated covariance matrix $\hat{\boldsymbol{\Sigma}}(\mathbf{x}_t)$ as a diagonal matrix with constant values $\beta_t$ or $\tilde{\beta}_t$, and calculate the estimated mean $\hat{\mu}(\mathbf{x}_t)$ as follows,

$$\hat{\mu}(\mathbf{x}_t) = \frac{1}{\sqrt{\alpha_t}}(\mathbf{x}_t - \frac{\beta_t}{\sqrt{1-\bar{\alpha}_t}}\hat{\epsilon}(\mathbf{x}_t)) \tag{2}$$

where $\hat{\epsilon}(\mathbf{x}_t)$ is the estimated noise at time $t$.

### 3.2 Multinomial Diffusion Kernel

The multinomial diffusion kernel is applied in the forward process for discrete features with the same noise schedule $\beta_t$ used by the Gaussian diffusion kernel. For discrete features, in the final timestep $T$, $\mathbf{x}_T$ follows the discrete uniform distribution Uniform$(K)$, where $K$ is the number of categories. The multinomial diffusion kernel can be written as:

$$q(\mathbf{x}_t|\mathbf{x}_{t-1}) = \text{Cat}(\mathbf{x}_t|(1-\beta_t)\mathbf{x}_{t-1} + \beta_t/K)$$

Similarly to the continuous case, the posterior distribution of the multinomial diffusion kernel has an analytical solution.

$$q(\mathbf{x}_{t-1}|\mathbf{x}_t, \mathbf{x}_0) = \text{Cat}(\mathbf{x}_{t-1}|\tilde{\theta}/\sum_{k=1}^{K}\tilde{\theta}_k)$$

$$\tilde{\theta} = [\alpha_t\mathbf{x}_t + (1-\alpha_t)/K] \odot [\bar{\alpha}_{t-1}\mathbf{x}_0 + (1-\bar{\alpha}_{t-1})/K]$$

The posterior estimator for discrete features is preferably parameterized by the estimated starting point $\hat{\mathbf{x}}_0(\mathbf{x}_t)$ as suggested by Hoogeboom et al. (2021).

$$p_\theta(\mathbf{x}_{t-1}|\mathbf{x}_t) = q(\mathbf{x}_{t-1}|\mathbf{x}_t, \hat{\mathbf{x}}_0(\mathbf{x}_t)) \tag{3}$$

### 3.3 Model Fitting

The parameters $\theta$ of the posterior estimator are learned by minimizing the variational lower bound,

$$\mathcal{L}(\mathbf{x}_0) = \mathbb{E}_{q(\mathbf{x}_0)}[D(q(\mathbf{x}_T|\mathbf{x}_0)\|p(\mathbf{x}_T)) - \log p_\theta(\mathbf{x}_0|\mathbf{x}_1)$$
$$+ \sum_{t=2}^{T} D(q(\mathbf{x}_{t-1}|\mathbf{x}_t, \mathbf{x}_0)\|p_\theta(\mathbf{x}_{t-1}|\mathbf{x}_t))] \tag{4}$$

where Kullback–Leibler divergence is denoted by $D$. The variational lower bound can be simplified for continuous variables as the mean squared error between true noise $\epsilon$ in Equation (1) and estimated noise $\hat{\epsilon}$ in Equation (2), denoted by $\mathcal{L}_{\text{G}}$.

$$\mathcal{L}_{\text{G}} = \mathbb{E}_{q(\mathbf{x}_0)}[\|\epsilon - \hat{\epsilon}(\mathbf{x}_t)\|^2]$$

For mixed-type data containing continuous features and $C$ categorical features, the total loss $\mathcal{L}_{\text{T}}$ can be expressed as the summation of the Gaussian diffusion loss term $\mathcal{L}_{\text{G}}$ and the average of the multinomial diffusion loss terms in Equation (4) for all categorical features:

$$\mathcal{L}_{\text{T}} = \mathcal{L}_{\text{G}} + \frac{\sum_{i \le C}\mathcal{L}^{(i)}}{C}$$

To train latent diffusion models, let $\mathbf{z}_t$ be the latent representation of $\mathbf{x}_t$, the estimated noise $\hat{\epsilon}(\mathbf{x}_t)$ for continuous features in Equation (2) can be reformulated as $\hat{\epsilon}(\mathbf{z}_t)$. The estimated original input $\hat{\mathbf{x}}_0(\mathbf{x}_t)$ in Equation (3) can be reformulated as $\hat{\mathbf{x}}_0(\mathbf{z}_t)$.

### 3.4 Classifier-Free Guidance

Classifier-free guidance (Ho & Salimans, 2022) is a mechanism used in diffusion models to condition the generation process on auxiliary information without requiring an external classifier. Below, we provide a mathematical overview of this method. Let $x_t$ denote the noisy data at timestep $t$ in the reverse diffusion process, and let $c$ represent the conditional information (e.g., a class label). The reverse diffusion process aims to estimate the posterior $p(x_{t-1}|x_t, c)$, which is parameterized by a neural network $\epsilon_\theta$. Classifier-free guidance interpolates between conditional and unconditional estimates of the noise, as shown below,

$$\epsilon(x_t, c) = \epsilon_\theta(x_t) + w_g(\epsilon_\theta(x_t, c) - \epsilon_\theta(x_t))$$

where $\epsilon_\theta(x_t)$ is the unconditional noise estimate, $\epsilon_\theta(x_t, c)$ is the conditional noise estimate, and $w_g$ is the guidance weight, controlling the influence of the conditional term. The term $\epsilon_\theta(x_t, c) - \epsilon_\theta(x_t)$ represents the guidance signal derived from the condition $c$. By scaling this signal with $w_g$, the model adjusts the strength of conditioning during generation. Larger values of $w_g$ enforce stronger adherence to the condition, while smaller values reduce its influence, yielding outputs closer to the unconditional distribution.

For tasks involving multiple conditions, the guidance term can be extended to support multivariate conditioning. Suppose we have $N$ conditions $c_1, c_2, \ldots, c_N$, the guided noise estimate can be formulated as,

$$\epsilon(x_t, c_1, c_2, \ldots, c_N) = \epsilon_\theta(x_t) + \sum_{i=1}^{N} w_{g_i} \gamma(x_t, c_i)$$

where $\gamma(x_t, c_i) = \epsilon_\theta(x_t, c_i) - \epsilon_\theta(x_t)$ represents the guidance signal for the $i$-th condition, and $w_{g_i}$ represents the $i$-th guidance weight. The overall guidance is computed as the sum of these individual contributions across all conditions.

## 4 Methods

As explained in Section 1, generative diffusion models for tabular data are prone to learning the sensitive group size disparity in training data. In this section, we describe the process of synthesizing balanced mixed-type tabular data considering sensitive attributes. We first introduce conditional generation given multivariate guidance while preserving the synthesis quality. Then, we explain the architecture of the backbone deep neural network used as the posterior estimator in our method. Lastly, we explain the procedure for sampling balanced data with consideration for sensitive attributes. The high-level structure of our method is shown in Figure 1.

### 4.1 Multivariate Latent Guidance

To achieve conditional data generation with diffusion models, classifier-free guidance (Ho & Salimans, 2022) is a simple but effective approach. Intuitively, the classifier-free guidance method tries to let the posterior estimator "know" the label of the data it models. In latent space, it uses one neural network to receive $\mathbf{z}_t$ and condition $\mathbf{c}$ to make estimations. We can denote the estimated noise given corresponding condition $\mathbf{c}$ for continuous features as $\bar{\epsilon}(\mathbf{z}_t, \mathbf{c})$,

$$\bar{\epsilon}(\mathbf{z}_t, \mathbf{c}) = \hat{\epsilon}(\mathbf{z}_t) + w_g(\underbrace{\hat{\epsilon}(\mathbf{z}_t, \mathbf{c}) - \hat{\epsilon}(\mathbf{z}_t)}_{\gamma(\mathbf{z}_t, \mathbf{c})}) \tag{5}$$

where $\gamma(\mathbf{z}_t, \mathbf{c}) = \hat{\epsilon}(\mathbf{z}_t, \mathbf{c}) - \hat{\epsilon}(\mathbf{z}_t)$ is guidance of $\mathbf{c}$ and it is scaled by guidance weight $w_g$.

Equation (5) can be extended to support multivariate guidance. Especially in tabular synthesis, we want to generate samples conditioned on labels $\mathbf{c}$ and sensitive features $\mathbf{s}$. In this case, we can let our neural network take $\mathbf{z}_t$, $\mathbf{c}$, and $\mathbf{s}$ as inputs, resulting in:

$$\bar{\epsilon}(\mathbf{z}_t, \mathbf{c}, \mathbf{s}) = \hat{\epsilon}(\mathbf{z}_t) + w_g(\gamma(\mathbf{z}_t, \mathbf{c}) + \gamma(\mathbf{z}_t, \mathbf{c}, \mathbf{s})) \tag{6}$$

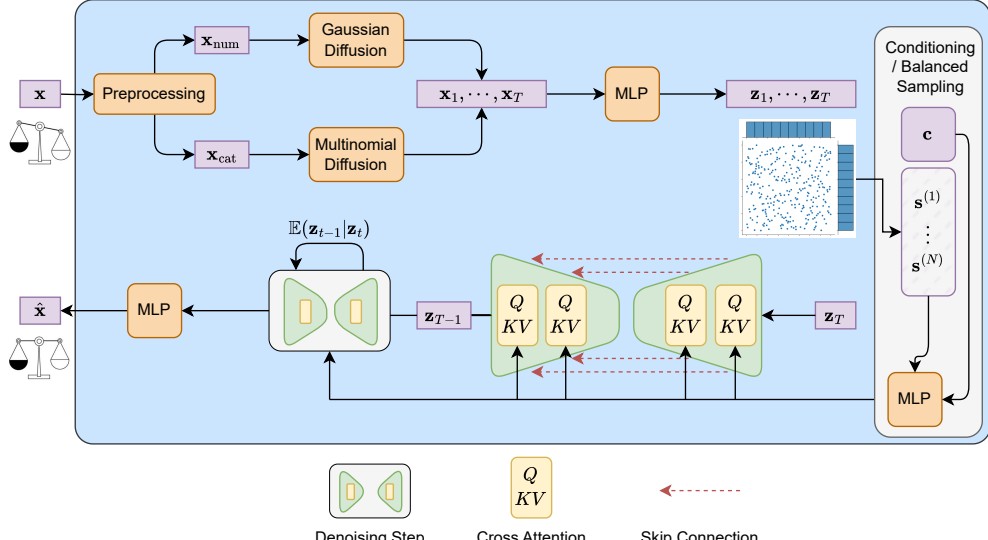

Figure 1: The diagram of our model architecture. In the forward process, the input data point $\mathbf{x}$ is pre-processed into numerical part $\mathbf{x}_{\text{num}}$ and categorical part $\mathbf{x}_{\text{cat}}$, and then passing through $T$ steps of the diffusion kernel to get $\mathbf{x}_1, \cdots, \mathbf{x}_T$. $\mathbf{z}_1, \cdots, \mathbf{z}_T$ is the latent representation of $\mathbf{x}_1, \cdots, \mathbf{x}_T$. In the reverse process, the posterior estimator iteratively denoises noisy input $\mathbf{z}_T$ conditioning on an outcome $\mathbf{c}$ and $N$ sensitive attributes $\mathbf{s}^{(1)}, \cdots, \mathbf{s}^{(N)}$. The estimated data point is $\hat{\mathbf{x}}$. Our model can generate fair synthetic data by leveraging sensitive guidance to ensure a balanced joint distribution of the target label and sensitive attributes.

Using the mechanism of multivariate classifier-free guidance introduced in Section 3.4, Equation (6) extends label-only conditioning in classifier-free guidance with multivariate feature-level conditioning. The term $\gamma(\mathbf{z}_t, \mathbf{c})$ is the guidance term for the target label, representing the adjustment to the noise estimate based on the label condition. The term $\gamma(\mathbf{z}_t, \mathbf{c}, \mathbf{s})$ is the sensitive guidance term, incorporating adjustments based on the sensitive attribute $\mathbf{s}$, and the label condition $\mathbf{c}$ here is used as a reference to restrict the influence of the sensitive guidance, ensuring fairness without distorting the primary label distribution. Since the primary conditional guidance is the target label, incorporating sensitive guidance from multiple attributes should influence the synthetic data without significantly altering it compared to using only the target label as guidance. To achieve this, the sensitive guidance $\gamma(\mathbf{z}_t, \mathbf{c}, \mathbf{s})$ is controlled element-wisely by a "security gate" $\mu(\mathbf{c}, \mathbf{s}; w_s, \lambda)$,

$$\gamma(\mathbf{z}_t, \mathbf{c}, \mathbf{s}) = \mu(\mathbf{c}, \mathbf{s}; w_s, \lambda)(\hat{\epsilon}(\mathbf{z}_t, \mathbf{s}) - \hat{\epsilon}(\mathbf{z}_t))$$

$$\mu(\mathbf{c}, \mathbf{s}; w_s, \lambda) = \begin{cases} \phi, & \text{where } \hat{\epsilon}(\mathbf{z}_t, \mathbf{c}) \ominus \hat{\epsilon}(\mathbf{z}_t, \mathbf{s}) < \lambda \\ 0, & \text{otherwise} \end{cases}$$

$$\phi = \max(1, w_s|\hat{\epsilon}(\mathbf{z}_t, \mathbf{c}) - \hat{\epsilon}(\mathbf{z}_t, \mathbf{s})|)$$

where $\ominus$ is element-wise subtraction, and $w_s$ is the sensitive guidance weight. The purpose of $\mu(\mathbf{c}, \mathbf{s}; w_s, \lambda)$ is to keep the alterations caused by the sensitive guidance steady and minimal. It deactivates the sensitive guidance when the difference between the estimation conditioned on $\mathbf{c}$ and the estimation conditioned on $\mathbf{s}$ is larger than a threshold $\lambda$. Furthermore, sensitive guidance is deactivated in the early phase of the diffusion process when $t$ is smaller than a warm-up timestep $\delta$ (a positive integer indicating the timestep at which sensitive guidance starts):

$$\gamma(\mathbf{z}_t, \mathbf{c}, \mathbf{s}) := \mathbf{0} \text{ if } t < \delta$$

Additionally, a momentum term with momentum weight $w_m$ is added to the sensitivity guidance to make guidance direction stable,

$$\gamma_t(\mathbf{z}_t, \mathbf{c}, \mathbf{s}) = \mu(\mathbf{c}, \mathbf{s}; w_s, \lambda)(\hat{\epsilon}(\mathbf{z}_t, \mathbf{s}) - \hat{\epsilon}(\mathbf{z}_t)) + w_m \nu_t$$

$$\nu_{t+1} = \beta \nu_t + (1 - \beta)\gamma_t$$

where $\nu_0 = \mathbf{0}$ and $\beta \in [0, 1]$. To include not only one sensitive feature, but $N$ sensitive features $\mathbf{S} = \{\mathbf{s}^{(1)}, \cdots, \mathbf{s}^{(N)}\}$, we can extend the sensitive guidance as follows:

$$\gamma(\mathbf{z}_t, \mathbf{c}, \mathbf{S}) = \sum_{i=1}^{N} \mu(\mathbf{c}, \mathbf{s}^{(i)}; w_s, \lambda)(\hat{\epsilon}(\mathbf{z}_t, \mathbf{s}^{(i)}) - \hat{\epsilon}(\mathbf{z}_t))$$

By this approach, we can model continuous features given labels $\mathbf{c}$ and a set of sensitive features $\mathbf{S}$. Similarly, the estimated starting point given $\mathbf{c}$ and $\mathbf{S}$ for discrete features can be formulated as follows,

$$\bar{\mathbf{x}}_0(\mathbf{z}_t, \mathbf{c}, \mathbf{S}) = \hat{\mathbf{x}}_0(\mathbf{z}_t) + w_g(\gamma(\mathbf{z}_t, \mathbf{c}) + \gamma(\mathbf{z}_t, \mathbf{c}, \mathbf{S})) \tag{7}$$

with $\gamma(\mathbf{z}_t, \mathbf{c})$ and $\gamma(\mathbf{z}_t, \mathbf{c}, \mathbf{S})$ parameterized by $\hat{\mathbf{x}}_0$.

## 4.2  Backbone

In this research, we model mixed-type tabular data in latent space following a similar setup in (Rombach et al., 2022). The tabular data are encoded into latent space and decoded back with multilayer perceptrons. We choose U-Net (Ronneberger et al., 2015) with transformers as a posterior estimator to predict $\bar{\epsilon}(\mathbf{z}_t, \mathbf{c}, \mathbf{S})$ and $\bar{\mathbf{x}}_0(\mathbf{z}_t, \mathbf{c}, \mathbf{S})$. This choice is based on the hypothesis that the U-Net, augmented by the contextual understanding capabilities of transformers, can effectively manage the heterogeneous nature of tabular data. This integration leverages the U-Net's ability to capture spatial correlation and the transformers' strengths in sequence modeling, thereby offering a novel approach to latent space modeling of tabular data.

## 4.3  Balanced Sampling

After training diffusion models to approximate the distribution of the training data conditioned on labels $\mathbf{c}$ and sensitive features $\mathbf{S}$, we can generate samples from this conditional distribution and align them with any desired target distribution by adjusting the conditioning variables. Specifically, during sampling, the denoising neural network utilizes guidance from the target label and sensitive attributes. The joint distribution of the target label and sensitive attributes can be customized. In our bias mitigation framework, we expect the synthetic data to preserve the same label distribution as the real data, while making the sensitive attributes nearly independent of the target label. To achieve this, we first compute the empirical label distribution from the real data and apply uniform categorical distributions for each sensitive feature. This results in a balanced joint distribution of the target label and sensitive features, which serves as input for the conditional generation in our diffusion model. Then our diffusion model iteratively denoises the Gaussian noise with label and sensitive guidance to generate fair tabular data.

## 5  Experiments

In this section, we first introduce experimental datasets, data processing, baselines, and how we assess synthetic tabular data. Then we present computational results on experimental datasets. The training settings are presented in Appendix B.

## 5.1  Datasets

We evaluate the effectiveness and fairness of our proposed method using three binary classification tabular datasets: Adult (Kohavi et al., 1996), Bank Marketing (Moro et al., 2014), and COMPAS (Larson et al., 2016). These datasets contain numerical and categorical features, including some sensitive attributes with

observed class imbalances or strong correlations to the target variable. We provide a summary of the datasets, including their corresponding sensitive attributes in Table 1, and offer further details in Appendix A. Common sensitive attributes such as sex and race can be found in Adult and COMPAS. As mentioned in (Rajabi & Garibay, 2022), the age group in the Bank Marketing dataset is considered sensitive, with individuals below 25 classified as "young" and those above 25 classified as "old". In our experiments, each data set is divided into training, validation, and test sets using a 50/25/25 split. Following the training of our diffusion models with multivariate guidance, we proceed to generate synthetic data that adheres to the original label distribution while incorporating a uniform distribution for sensitive features.

Table 1: Experimental datasets.

| Dataset | Train | Validation | Test | Sensitive | Target |
|---------|-------|------------|------|-----------|--------|
| Adult | 22611 | 11305 | 11306 | Sex & Race | Income |
| Bank | 22605 | 11303 | 11303 | Age | Subscription |
| COMPAS | 8322 | 4161 | 4161 | Sex & Race | Recidivism |

## 5.2 Data Processing and Baselines

### 5.2.1 Data Processing

Following the data processing steps in Kotelnikov et al. (2023), we split the numerical and categorical features for each dataset and pre-process them separately. Numerical features are converted through quantile transformation, a non-parametric technique transforming variables into a standard Gaussian distribution based on quantiles, thus normalizing the data and enhancing the robustness to outliers. In preparation for the Markov transition process in the latent diffusion model, we encode categorical features into one-hot vectors and concatenate the resulting vectors with normalized numerical features.

### 5.2.2 Baselines

To evaluate the effectiveness of our approach across different datasets, we perform a comparative analysis against state-of-the-art (SOTA) baseline models that have publicly available code for tabular data synthesis. Additionally, we compare our method with fair tabular generative models based on GAN and SMOTE. The specific baseline methods chosen for comparison are detailed below.

- **Fair Class Balancing (FairCB)** (Yan et al., 2020): a SMOTE-based oversampling method to adjust class distributions in the training data to address class imbalances.

- **TabFairGAN (FairTGAN)** (Rajabi & Garibay, 2022): a fairness-aware GAN tailored for tabular data incorporates a fair discriminator to ensure equality in the joint probability distribution of sensitive features and labels.

- **CoDi** (Lee et al., 2023): a recent diffusion-based approach that combines two diffusion models: one for continuous variables and another for discrete variables. These models are trained separately to effectively learn the distributions of their respective data types. However, they are also linked together by conditioning each model's sample generation on the other model's samples at each time step, which enhances the learning of correlations between continuous and discrete variables. Additionally, to further strengthen the connection between the two models, contrastive learning is applied to both diffusion models. Negative samples are created by breaking the inter-correlation between the continuous and discrete variable sets, thus reinforcing the models' ability to learn these relationships.

- **GReaT** (Borisov et al., 2022): utilizes pre-trained transformer-decoder language models (LLMs) to generate synthetic tabular data through a two-stage process. In the first stage, it transforms each row of the real tabular dataset into a textual representation by representing each feature with its name and value in a subject-predicate-object structure. The transformation of these features is then concatenated in a random order to form the feature vector of the corresponding row. The random

ordering enables the LLM to be fine-tuned on samples without depending on the sequence of features, allowing for arbitrary conditioning during tabular data generation. This textual representation of the tabular dataset is used to fine-tune the pre-trained LLM. In the second stage, the fine-tuned LLM generates synthetic data by initializing it with any combination of feature names or feature name-value pairs, allowing the model to complete the remaining feature vector in its textual representation, which can then be converted back into tabular format.

- **SMOTE** (Chawla et al., 2002): a robust interpolation-based method that creates synthetic data points by combining a real data point with its $k$ nearest neighbors from the dataset. In our application, we leverage SMOTE to generate additional tabular data samples, employing interpolation among existing samples with the same label. The implementation of SMOTE is available in Lemaître et al. (2017).

- **STaSy** (Kim et al., 2022): a Score-based Tabular Data Synthesis (STaSy) method that adopts the score-based generative modeling approach. It introduces a self-paced learning method that begins by training on simpler records with a denoising score matching loss below a certain threshold. As training progresses and the model becomes more robust, progressively more challenging records are included in the training. Additionally, STaSy includes a fine-tuning approach that adjusts the model parameters by iteratively retraining it using the log probabilities of the generated samples.

- **TabDDPM** (Kotelnikov et al., 2023): a recent diffusion-based model that has demonstrated to surpass existing methods for synthesizing tabular data. Leveraging multilayer perceptrons as its backbone, TabDDPM excels in learning unbalanced distributions in the training data.

- **TabSyn** (Zhang et al., 2023): a SOTA method that utilizes a latent diffusion model to generate tabular data containing both numerical and categorical features. First, a VAE model, specifically designed for tabular-structured data, converts raw tabular data with both numerical and categorical features into embeddings in a regularized latent space that has a distribution close to the standard normal distribution. Following this, a score-based diffusion model is trained in the embedding space to capture the distribution of these latent embeddings. The diffusion model adds Gaussian noise that increases linearly over time in the forward process, which minimizes approximation errors in the reverse process, thus allowing for synthetic data to be generated in significantly fewer reverse steps.

### 5.3 Assessing Synthetic Tabular Data

There are two primary approaches to evaluating synthetic tabular data. One approach is a model-based method that involves training machine learning models on synthetic tabular data and evaluating the trained models on validation data extracted from the original dataset. The other approach is a data-based method that operates independently of machine learning models and directly compares synthetic and real tabular data.

#### 5.3.1 Data-Based Evaluation

We evaluate synthetic tabular data by directly comparing it with real data, with a focus on column-wise density estimation and pair-wise column correlations. To address privacy concerns when sharing tabular data publicly, synthetic data should not replicate the original data. As suggested by (Zhang et al., 2023), we compute **"distance to the closest record" (DCR)** scores to ensure this. Furthermore, when assessing the fairness of synthetic tabular data using data-based methods, we analyze the class imbalances within sensitive features like sex or race, as well as the joint distribution with the target label.

#### 5.3.2 Model-Based Evaluation

The model-based evaluation process for synthetic tabular data in classification tasks involves the following steps: first, training a generative model on the original training set; next, generating synthetic data using the trained generative model; then, training classifiers on the synthetic data; and finally, evaluating these

classifiers on the original test data. Specifically, when assessing model accuracy on the original test data, it is referred to as **machine learning efficiency** as used in (Choi et al., 2017).

Beyond accuracy assessments, the evaluation of synthetic data can extend to fairness scores of classifiers, providing insights into the fairness of the generated data. We evaluate the fairness of machine learning models trained on the synthetic data, focusing on two key aspects: equal allocation and equal performance (Agarwal et al., 2018).

- Equal allocation, a fundamental principle in model-based fairness evaluations, entails the idea that a model should distribute resources or opportunities proportionally among different groups, irrespective of their affiliation with any privileged group. Equal allocation is quantified using the **demographic parity ratio (DPR)**. This ratio reveals the extent of the imbalance by comparing the lowest and highest selection rates (the proportion of examples predicted as positive) between groups.

- Equal performance is grounded in the principle that a fair model should exhibit consistent performance across all groups. This entails maintaining the same precision level for each group. The **equalized odds ratio (EOR)**, which represents the smaller of two ratios comparing true and false positive rates between groups, serves as a metric for assessing performance fairness in this context.

Both the demographic parity ratio and equalized odds ratio are within the range $[0, 1]$. The demographic parity ratio highlights the importance of distributing resources equally among different groups, whereas the equalized odds ratio prioritizes impartial decision-making within each specific group. Higher values in either metric indicate progress toward fairness. In our experiments, when evaluating the fairness scores of classifiers as shown in Table 4, we evaluate the fairness scores of classifiers by selecting sex as the sensitive attribute for the Adult and COMPAS datasets, and age group for the Bank dataset.

## 5.4 Computational Results

This section presents the computational results of our evaluation of synthetic tabular data generated with three different random seeds. We trained the SOTA machine learning model for tabular data, CatBoost, on three different sets of synthetic data, each generated with a unique random seed.

### 5.4.1 Fidelity and Diversity of Synthetic Data

Table 2: The values of error rates of column-wise density estimation and pair-wise column correlations.

| | Adult | Bank | COMPAS | Mean | | Adult | Bank | COMPAS | Mean |
|---|---|---|---|---|---|---|---|---|---|
| | **Density ($\downarrow$)** | | | % | | **Correlation ($\downarrow$)** | | | % |
| CoDi | $.145_{\pm.000}$ | $.154_{\pm.000}$ | $.205_{\pm.001}$ | 16.8% | | $.495_{\pm.000}$ | $.344_{\pm.001}$ | $.550_{\pm.001}$ | 46.3% |
| GReaT | $.077_{\pm.000}$ | $.102_{\pm.001}$ | $.093_{\pm.001}$ | 9.1% | | $.208_{\pm.015}$ | $.213_{\pm.010}$ | $.165_{\pm.016}$ | 19.5% |
| SMOTE | $.025_{\pm.000}$ | $.020_{\pm.000}$ | $.021_{\pm.001}$ | 2.2% | | $.054_{\pm.004}$ | $.042_{\pm.005}$ | $.047_{\pm.005}$ | 4.8% |
| STaSy | $.102_{\pm.000}$ | $.182_{\pm.000}$ | $.108_{\pm.001}$ | 13.1% | | $.163_{\pm.000}$ | $.221_{\pm.001}$ | $.138_{\pm.001}$ | 17.4% |
| TabDDPM | $.037_{\pm.001}$ | $.028_{\pm.001}$ | $.057_{\pm.001}$ | 4.1% | | $.055_{\pm.001}$ | $.052_{\pm.001}$ | $.090_{\pm.001}$ | 6.6% |
| TabSyn | $.010_{\pm.000}$ | $.009_{\pm.000}$ | $.027_{\pm.001}$ | 1.5% | | $.035_{\pm.001}$ | $.033_{\pm.002}$ | $.054_{\pm.001}$ | 4.1% |
| FairCB | $.076_{\pm.000}$ | $.066_{\pm.000}$ | $.039_{\pm.000}$ | 6.0% | | $.125_{\pm.000}$ | $.111_{\pm.000}$ | $.074_{\pm.000}$ | 10.3% |
| FairTGAN | $.034_{\pm.000}$ | $.030_{\pm.000}$ | $.055_{\pm.001}$ | 4.0% | | $.080_{\pm.001}$ | $.053_{\pm.000}$ | $.087_{\pm.001}$ | 7.3% |
| Ours | $.126_{\pm.001}$ | $.121_{\pm.000}$ | $.109_{\pm.001}$ | 11.9% | | $.201_{\pm.000}$ | $.174_{\pm.005}$ | $.174_{\pm.003}$ | 18.3% |

Table 2 and Table 3 show that our method remains competitive with SOTA models in terms of column-wise density estimation, pair-wise column correlations, and machine learning efficiency, even after incorporating a uniformly distributed direction for sensitive features during the sampling phase. Across the three experimental datasets, our model achieved an average AUC of 84.7%, which is only 1.3% lower than TabSyn, the SOTA deep learning tabular synthesis method. Although altering the original sensitive distribution is anticipated to reduce AUC, the results demonstrate that our method effectively preserves high fidelity in the generated synthetic data. While SMOTE and FairCB excel in fidelity, their low DCR scores suggest

Table 3: The values of machine learning efficiency with the best classifier for each dataset and DCR scores.

| | Adult | Bank | COMPAS | Mean | Adult | Bank | COMPAS | Mean |
|---|---|---|---|---|---|---|---|---|
| | **AUC (↑)** | | | % | **DCR Scores (Ideally Around .500)** | | | % |
| Real | $.928_{\pm.000}$ | $.936_{\pm.000}$ | $.810_{\pm.001}$ | 89.1% | - | - | - | - |
| CoDi | $.858_{\pm.001}$ | $.826_{\pm.014}$ | $.678_{\pm.002}$ | 78.7% | $.331_{\pm.003}$ | $.348_{\pm.001}$ | $.400_{\pm.002}$ | 36.0% |
| GReaT | $.901_{\pm.002}$ | $.688_{\pm.024}$ | $.717_{\pm.008}$ | 76.9% | $.320_{\pm.002}$ | $.348_{\pm.003}$ | $.377_{\pm.003}$ | 34.8% |
| SMOTE | $.914_{\pm.000}$ | $.928_{\pm.001}$ | $.778_{\pm.002}$ | 87.3% | $.327_{\pm.004}$ | $.265_{\pm.001}$ | $.273_{\pm.003}$ | 28.8% |
| STaSy | $.885_{\pm.003}$ | $.895_{\pm.003}$ | $.728_{\pm.013}$ | 83.6% | $.344_{\pm.001}$ | $.345_{\pm.002}$ | $.362_{\pm.001}$ | 35.0% |
| TabDDPM | $.907_{\pm.001}$ | $.917_{\pm.002}$ | $.745_{\pm.001}$ | 85.6% | $.339_{\pm.000}$ | $.350_{\pm.002}$ | $.367_{\pm.005}$ | 35.2% |
| TabSyn | $.911_{\pm.000}$ | $.919_{\pm.000}$ | $.749_{\pm.001}$ | 86.0% | $.339_{\pm.002}$ | $.351_{\pm.005}$ | $.367_{\pm.006}$ | 35.2% |
| FairCB | $.915_{\pm.001}$ | $.907_{\pm.002}$ | $.771_{\pm.001}$ | 86.4% | $.054_{\pm.000}$ | $.031_{\pm.001}$ | $.012_{\pm.001}$ | 3.2% |
| FairTGAN | $.881_{\pm.000}$ | $.863_{\pm.006}$ | $.705_{\pm.002}$ | 81.6% | $.348_{\pm.002}$ | $.348_{\pm.004}$ | $.374_{\pm.006}$ | 35.7% |
| Ours | $.893_{\pm.002}$ | $.914_{\pm.002}$ | $.734_{\pm.001}$ | 84.7% | $.344_{\pm.001}$ | $.350_{\pm.002}$ | $.370_{\pm.004}$ | 35.5% |

that they closely mimic the real data, which is expected given their direct interpolation of the original data points.

### 5.4.2 Fairness Scores in Classification Tasks

Table 4: The values of DPR and EOR with best classifier for each dataset.

| | Adult | Bank | COMPAS | Mean | Adult | Bank | COMPAS | Mean |
|---|---|---|---|---|---|---|---|---|
| | **DPR (↑)** | | | % | **EOR (↑)** | | | % |
| Real | $.309_{\pm.001}$ | $.402_{\pm.015}$ | $.675_{\pm.006}$ | 46.2% | $.193_{\pm.005}$ | $.367_{\pm.024}$ | $.645_{\pm.015}$ | 40.2% |
| CoDi | $.293_{\pm.034}$ | $.189_{\pm.054}$ | $.855_{\pm.025}$ | 44.6% | $.247_{\pm.042}$ | $.172_{\pm.063}$ | $.857_{\pm.031}$ | 42.5% |
| GReaT | $.249_{\pm.015}$ | $.572_{\pm.289}$ | $.624_{\pm.066}$ | 48.2% | $.155_{\pm.028}$ | $.380_{\pm.126}$ | $.543_{\pm.075}$ | 35.9% |
| SMOTE | $.321_{\pm.006}$ | $.405_{\pm.028}$ | $.648_{\pm.021}$ | 45.8% | $.254_{\pm.011}$ | $.381_{\pm.026}$ | $.589_{\pm.009}$ | 40.8% |
| STaSy | $.261_{\pm.045}$ | $.468_{\pm.123}$ | $.436_{\pm.092}$ | 38.8% | $.182_{\pm.069}$ | $.451_{\pm.113}$ | $.433_{\pm.140}$ | 35.5% |
| TabDDPM | $.261_{\pm.006}$ | $.337_{\pm.020}$ | $.558_{\pm.041}$ | 38.5% | $.156_{\pm.007}$ | $.334_{\pm.028}$ | $.540_{\pm.047}$ | 34.3% |
| TabSyn | $.281_{\pm.017}$ | $.336_{\pm.016}$ | $.697_{\pm.040}$ | 43.8% | $.178_{\pm.019}$ | $.317_{\pm.023}$ | $.664_{\pm.062}$ | 38.6% |
| FairCB | $.286_{\pm.002}$ | $.719_{\pm.005}$ | $.675_{\pm.006}$ | 56.0% | $.192_{\pm.003}$ | $.801_{\pm.016}$ | $.638_{\pm.021}$ | 54.4% |
| FairTGAN | $.554_{\pm.006}$ | $.338_{\pm.093}$ | $.448_{\pm.025}$ | 44.7% | $.697_{\pm.017}$ | $.158_{\pm.033}$ | $.392_{\pm.041}$ | 41.6% |
| Ours | $.543_{\pm.026}$ | $.710_{\pm.057}$ | $.800_{\pm.080}$ | 68.4% | $.667_{\pm.059}$ | $.649_{\pm.040}$ | $.825_{\pm.113}$ | 71.4% |

As shown in Table 4, our method significantly outperforms all baseline models, achieving a mean DPR of 68.4% and a mean EOR of 71.4%. In comparison, state-of-the-art tabular generative models and FairTGAN fall below 50% for both fairness metrics, while FairCB remains below 60% for both. From Table 3, our method achieves an average AUC of 84.7 %, which is higher than FairTGAN and only 1.6 % lower than FairCB. By jointly optimizing performance and fairness, our method presents a more balanced trade-off than existing approaches. Moreover, in Appendix E.1, we demonstrate that simply overwriting the distribution of sensitive attributes is insufficient to mitigate bias in downstream classification tasks and can lead to a decrease in classification performance.

### 5.4.3 Sensitive Feature Distributions

In addition to evaluating the fairness of machine learning models, we also examine the class distribution of sensitive attributes in synthetic data. To achieve this, we use stacked bar plots to visualize the contingency tables for sensitive features and the target label. We further compare the distribution of sensitive attributes between real and synthetic data using bar plots.

We present the distribution of sensitive attributes versus the target label on the Adult dataset using our method in Figure 2. The synthetic data generated by our approach demonstrates a balanced joint distribution between sensitive features and the target label, effectively reducing the disparities observed in the original data. Additionally, we provide plots of the real data distribution and synthetic distributions generated by other methods in Appendix D. Note that we only visualize the synthetic data generated by our method, FairTGAN, and FairCB, which are three fairness-aware generative models, because we assume that the other baseline models replicate the real data distribution. In comparison, our method effectively produces a more

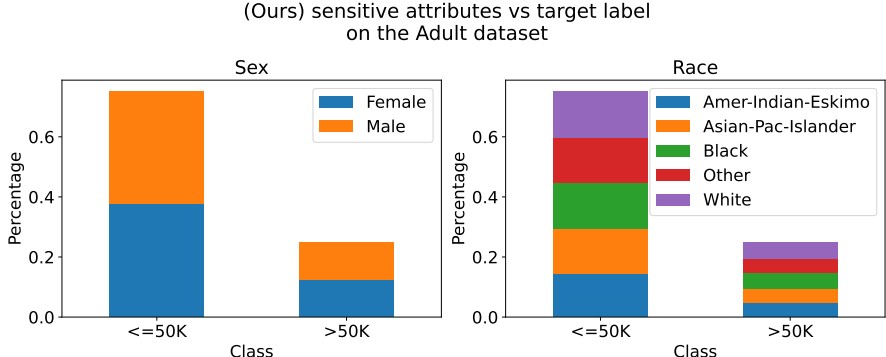

Figure 2: The distribution of sensitive attributes across different target label values on the Adult dataset using our method.

balanced joint distribution of sensitive attributes and target labels than the FairTGAN, FairCB, and the real distribution.

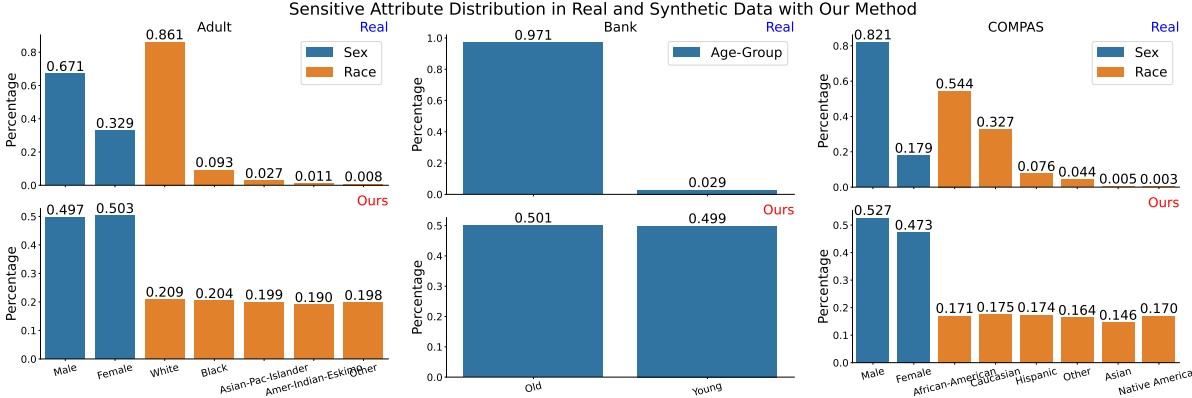

Figure 3: Comparison in the real versus synthetic distribution of sensitive attributes across all datasets.

### 5.4.4 Performance-Fairness Trade-Off

Balancing fairness and predictive accuracy is a key challenge in synthetic data generation, particularly when fairness improvements might impact important outcomes. This section explores how adjustments in fairness levels influence both performance and fairness metrics in the generated data. Our method provides a way to control this trade-off, allows practitioners to find a balance that meets their specific needs, whether they prioritize fairness, accuracy, or a compromise between the two.

To analyze the trade-off between fairness and accuracy, we define a balancing level $i$ ranging from 0 to 10. This level quantifies how much we adjust the original joint distribution of the target and sensitive attributes to make it more uniform.

Suppose we have $N$ sensitive attributes, each attribute with $C_i$ unique values, where $i$ ranges from 1 to $N$. This results in $K$ combinations of values where $K = C_i \times \cdots \times C_N$. Under the same targe label, each combination has a probability $y_k$ where $k$ is one of the $K$ combinations. For each combination, we calculate an offset $d_k = \bar{y} - y_k$, the difference between the average probability $\bar{y} = \sum_k y_k / |\mathbf{y}|$ and the current probability $y_k$. The offset $d_k$ represents how much the current joint probability deviation $y_k$ for a combination of sensitive attributes and the target label deviate from the average probability $\bar{y}$. By identifying these deviations, we can measure how far the joint distribution is from a uniform distribution, which is often desired for achieving

fairness. For a balancing level $i \in [0, 10]$, the balancing ratio is $i/10$, and the adjusted values $y_k^{\text{balanced}}$ are calculated as:

$$y_k^{\text{balanced}} = y_k + d_k \times \frac{i}{10} \tag{8}$$

The balanced values $y_k^{\text{balanced}}$ represent the adjusted probabilities for each combination of sensitive attributes and the target label after applying a fairness transformation. The balancing process ensures that $y_k^{\text{balanced}}$ systematically reduces disparities in the joint distribution of sensitive attributes and target labels, moving it closer to uniformity as the balancing level $i$ increases. By choosing an appropriate balancing level, the method allows for a practical balance between performance and fairness.

The results presented in the previous sections are based on a balancing level of 10, which achieves a nearly uniform joint distribution of the target and sensitive attributes. We conducted experiments to assess how different balancing levels affect the performance-fairness trade-off. We evaluated metrics such as AUC, DPR, and EOR, with DPR and EOR averaged across all sensitive attributes. We computed a composite score as a weighted sum of these metrics, with weights of 0.5 for AUC, 0.25 for DPR, and 0.25 for EOR. Figure 4 illustrates the trade-off between fairness and performance across different balancing levels. The optimal composite scores for the Adult and Bank datasets are achieved at a balancing level of 10, indicating a uniform distribution. For the COMPAS dataset, the best score is at level 9, representing a nearly uniform distribution. Practitioners can adjust the metric weights to suit specific application needs or establish thresholds for individual metrics based on their requirements.

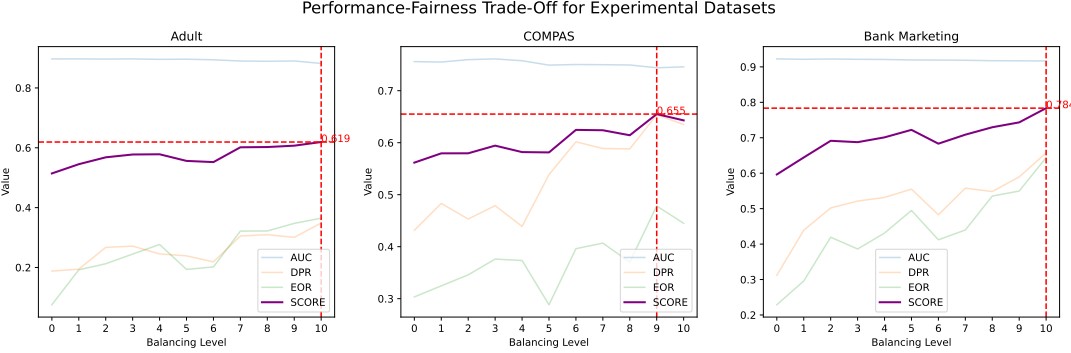

Figure 4: Trade-off between performance and fairness across balancing levels for experimental datasets. SCORE was computed as a weighted sum of these metrics, with weights of 0.5 for AUC, 0.25 for DPR, and 0.25 for EOR. The best composite scores for the Adult and Bank datasets are achieved at a balancing level of 10, while the COMPAS dataset achieves the best score at level 9.

## 6 Limitations and Discussion

Our proposed method is designed to generate balanced synthetic tabular data while considering sensitive attributes. However, it requires sensitive features to be specified in advance, which can be challenging in large-scale enterprise datasets with thousands of features. Furthermore, we could extend our testing to datasets with more than two sensitive attributes and imbalanced target distributions.

In terms of efficiency, we use a U-Net with attention layers as the backbone neural network for the posterior estimator in the diffusion framework, but this approach is more time-consuming compared to TabSyn. As the number of sensitive attributes increases, there are more hyperparameters to tune, which makes fine-tuning more computationally demanding, especially with the long sampling time. In the future, we aim to explore methods to reduce computational costs. We believe the main bottlenecks in computational costs are due to the long diffusion timesteps and the quadratic complexity of attention modules. Possible solutions include reducing diffusion timesteps via acceleration methods such as Flow-DPM-Solver Xie et al. (2024) and

refining the backbone, for instance, by employing linear attention Katharopoulos et al. (2020) or by using a pre-trained autoencoder to embed tabular data into a lower-dimensional latent space.

Lastly, our proposed method is designed to condition on specified group labels, which limits its applicability in scenarios where demographic information is unavailable or restricted due to privacy concerns. Future iterations of the model could integrate privacy-preserving techniques, such as synthetic attribute imputation, to enable fairness-aware data generation without requiring explicit sensitive attribute disclosure.

On the other hand, our proposed method generates fair data with a balanced joint distribution of the target label and multiple sensitive attributes, achieving higher fairness scores than baseline methods. Additionally, the model shows strong potential for extending to multimodal data synthesis. If we can improve the efficiency of our model and successfully adapt it to multimodal scenarios, we believe it could be widely applied to address fairness issues in fields such as healthcare, finance, and beyond.

## 7 Conclusion

In this work, we propose a novel diffusion model framework for mixed-type tabular data conditioned on both outcome and sensitive feature variables. Our approach leverages a multivariate guidance mechanism and performs balanced sampling considering sensitive features while ensuring a fair representation of the generated data. Extensive experiments on real-world datasets containing sensitive demographics demonstrate that our model achieves competitive performance and superior fairness compared to existing baselines.

## Impact Statements

The objective of this study is to make progress in the area of fair machine learning. Tabular datasets sometimes contain inherent bias, such as imbalanced distributions in sensitive attributes. Training machine learning models on biased datasets may result in decisions that could negatively affect minority groups. By mitigating biases present in tabular datasets through the generation of equitable synthetic data, our approach contributes to fostering equitable decision-making processes across industries such as finance, healthcare, and employment. Moreover, in instances where sharing datasets becomes necessary, the utilization of fair synthetic data ensures the preservation of user privacy and avoids causing emotional distress to minority groups. An adverse possibility is that bad individuals could manipulate distributions in sensitive attributes to generate biased (even stronger than original) data to harm minority groups. Additionally, the widespread release of synthetic data can diminish the quality of data sources available on the internet. Repeatedly training generative models on synthetic data in a self-consuming loop can lead to a decline in the quality of data synthesis, as discussed in (Alemohammad et al., 2023).

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

## A  Details about Datasets

### A.1  Adult

The Adult dataset (Kohavi et al., 1996) is widely used as a benchmark for exploring fairness and bias in machine learning. It contains 48842 data points. Each data point has 14 attributes and a binary target variable indicating whether an individual earns over fifty thousand dollars annually. The dataset encompasses employment, education, and demographic information, and sensitive features are sex and race. Download it from OpenML.

### A.2  Bank Marketing

The Bank Marketing dataset (Moro et al., 2014) is collected from direct marketing campaigns of a Portuguese banking institution. Comprising 45211 instances with 16 features, the predicted outcome is a binary variable indicating whether a client subscribed to a term deposit. Demographic, economic, and past marketing campaign data are included in features, and marital status is sensitive. Download it from OpenML.

### A.3  COMPAS

The COMPAS (Correctional Offender Management Profiling for Alternative Sanctions) dataset (Larson et al., 2016) is derived from a risk assessment tool used in the criminal justice system to evaluate the likelihood of recidivism among criminal defendants. It includes data on individuals arrested in Broward County, Florida, with 8 features and a binary outcome indicating whether the individual was predicted to reoffend. Features encompass demographic information, criminal history, and COMPAS risk scores, with sex and race being sensitive attributes. Download it from OpenML.

## B  Reproducibility

We performed our experiments on Ubuntu 20.04.2 LTS, utilizing Python version 3.10.14. Our framework of choice was PyTorch 2.3.0, with CUDA version 12.1, and we ran our computations on an NVIDIA GeForce RTX 3090. The hyperparameters we used for our method are listed as follows. The variable names can be found in our attached code. We tune the hyperparameters using Optuna Akiba et al. (2019) to optimize the validation AUC.

Table 5: The values of hyperparameters for our method.

| Notation | Variable | Meaning | Value/Search Space |
|---|---|---|---|
| - | `batch_size` | Batch size | 256 |
| $\delta$ | `warmup_steps` | Warm-up timestep | 0 |
| $\omega_s$ | `cond_guid_weight` | Sensitive guidance weight | 1.0 |
| $\lambda$ | `cond_guid_threshold` | Threshold of the "security gate" | 1.0 |
| $\omega_m$ | `cond_momentum_weight` | Momentum weight | 0.5 |
| $\beta$ | `cond_momentum_beta` | Momentum correction factor | 0.5 |
| $\omega_g$ | `overall_guid_weight` | Guidance weight | 1.0 |
| $T$ | `n_timesteps` | Number of timesteps in the diffusion process | $\{100, 1000\}$ |
| - | `n_epochs` | Number of training epochs | $\{100, 500, 1000\}$ |
| - | `lr` | Learning rate | $[0.00001, 0.003]$ |

For deep learning-based methods, we use the original hyperparameter settings provided in their respective implementations and mainly fine-tune the number of epochs, learning rate, and number of diffusion timesteps (if applicable) to optimize performance. For SMOTE-based methods, we adjust the number of nearest neighbors to achieve better results. We have detailed the hyperparameter search spaces for all baseline methods in Table 6. The hyperparameter search spaces are the same for all datasets.

Table 6: The values of hyperparameters for baseline methods.

| Method | Variable | Meaning | Value/Search Space |
|---|---|---|---|
| CoDi | `total_epochs_both`
`n_timesteps` | Number of training epochs
Number of timesteps in the diffusion process | $\{100, 500, 1000, 3000\}$
$\{100, 1000\}$ |
| GReaT | `batch_size`
`n_epochs` | Batch size
Number of training epochs | $\{4, 8\}$
$\{5, 10, 20\}$ |
| SMOTE | `knn` | Number of nearest neighbors for interpolation | $\{2, 21\}$ |
| STaSy | `lr`
`n_epochs` | Learning rate
Number of training epochs | $[0.00001, 0.003]$
$\{100, 500, 1000\}$ |
| TabDDPM | `lr`
`n_epochs`
`num_timesteps` | Learning rate
Number of training epochs
Number of timesteps in the diffusion process | $[0.00001, 0.003]$
$\{100, 500, 1000\}$
$\{100, 1000\}$ |
| TabSyn | `lr`
`n_epochs` | Learning rate
Number of training epochs | $[0.001, 0.002]$
$\{4000\}$ |
| FairCB | `knn` | Number of nearest neighbors for interpolation | $\{2, 21\}$ |
| FairTGAN | `lr`
`n_epochs`
`fair_epochs` | Learning rate
Number of training epochs
Number of fairness-related training epochs | $[0.00001, 0.003]$
$\{100, 500, 1000\}$
$\{100, 500\}$ |

## C  Efficiency

Table 7 provides a summary of the number of parameters, training time, and sampling time for various tabular data synthesis methods. Our proposed method is moderately sized, with 482,545 parameters, which is larger than TabDDPM but significantly smaller than most deep learning-based models, except FairTGAN. In terms of training efficiency, our method achieves a training time of 11.8 seconds per epoch, which is comparable to CoDi (10.5 seconds per epoch). However, the sampling time is a significant limitation, as it takes approximately 1,200 seconds to generate a synthetic dataset of the same size as the training data, making it substantially slower than all other baseline methods.

Table 7: Comparison of training and sampling efficiency across methods

| Method | Number of Parameters | Training Time (s/epoch) | Sampling Time (s) |
|---|---|---|---|
| CoDi | 2206711 | 10.5 | 9.1 |
| GReaT | 81912576 | 725.5 | 236.3 |
| SMOTE | - | 51.1 | 17.0 |
| STaSy | 10653326 | 1.4 | 25.1 |
| TabDDPM | 109672 | 6.0 | 118.9 |
| TabSyn | 10575912 | 0.6 | 2.7 |
| FairCB | - | 64.3 | 5.4 |
| FairTGAN | 45368 | 1.9 | 0.3 |
| Ours | 482545 | 11.8 | 1213.9 |

# D   Stacked Bar Plots of Contingency Tables

## D.1   Adult

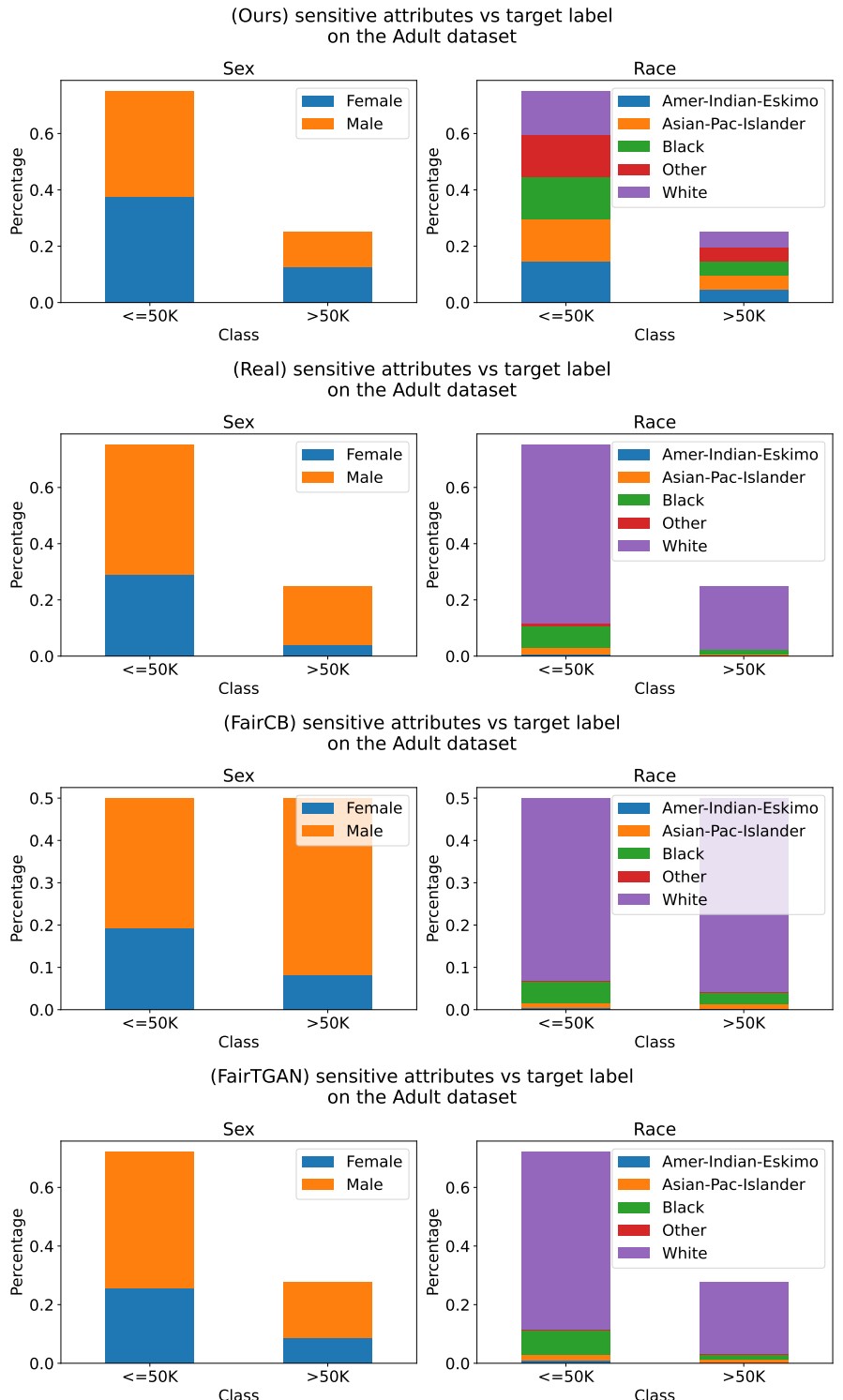

Figure 5: Comparison of models on the Adult dataset.

## D.2    Bank Marketing

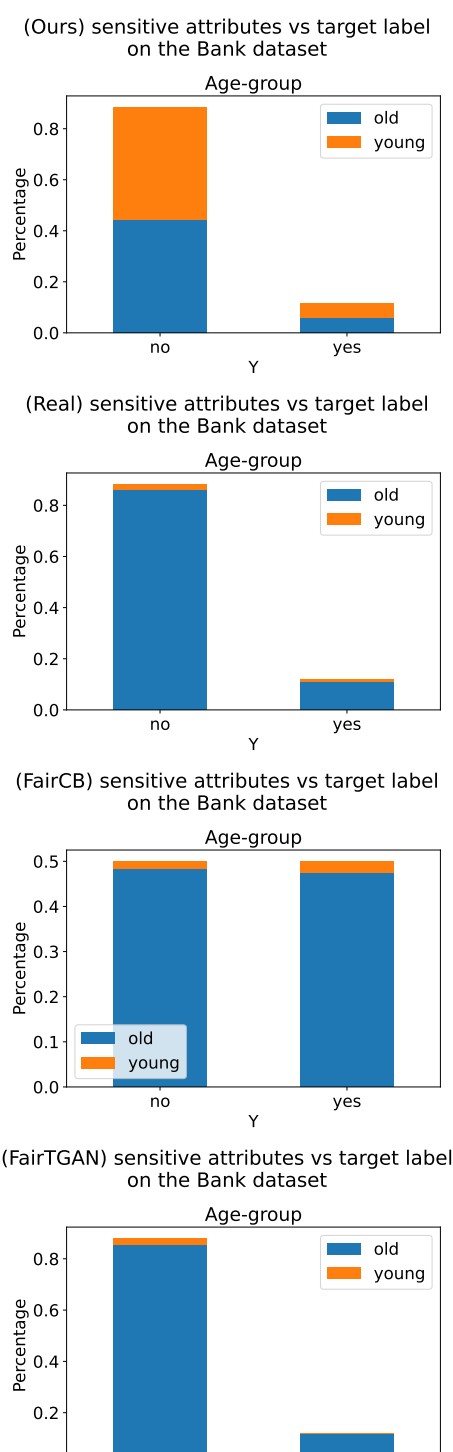

Figure 6: Comparison of models on the Bank Marketing dataset.

## D.3 COMPAS

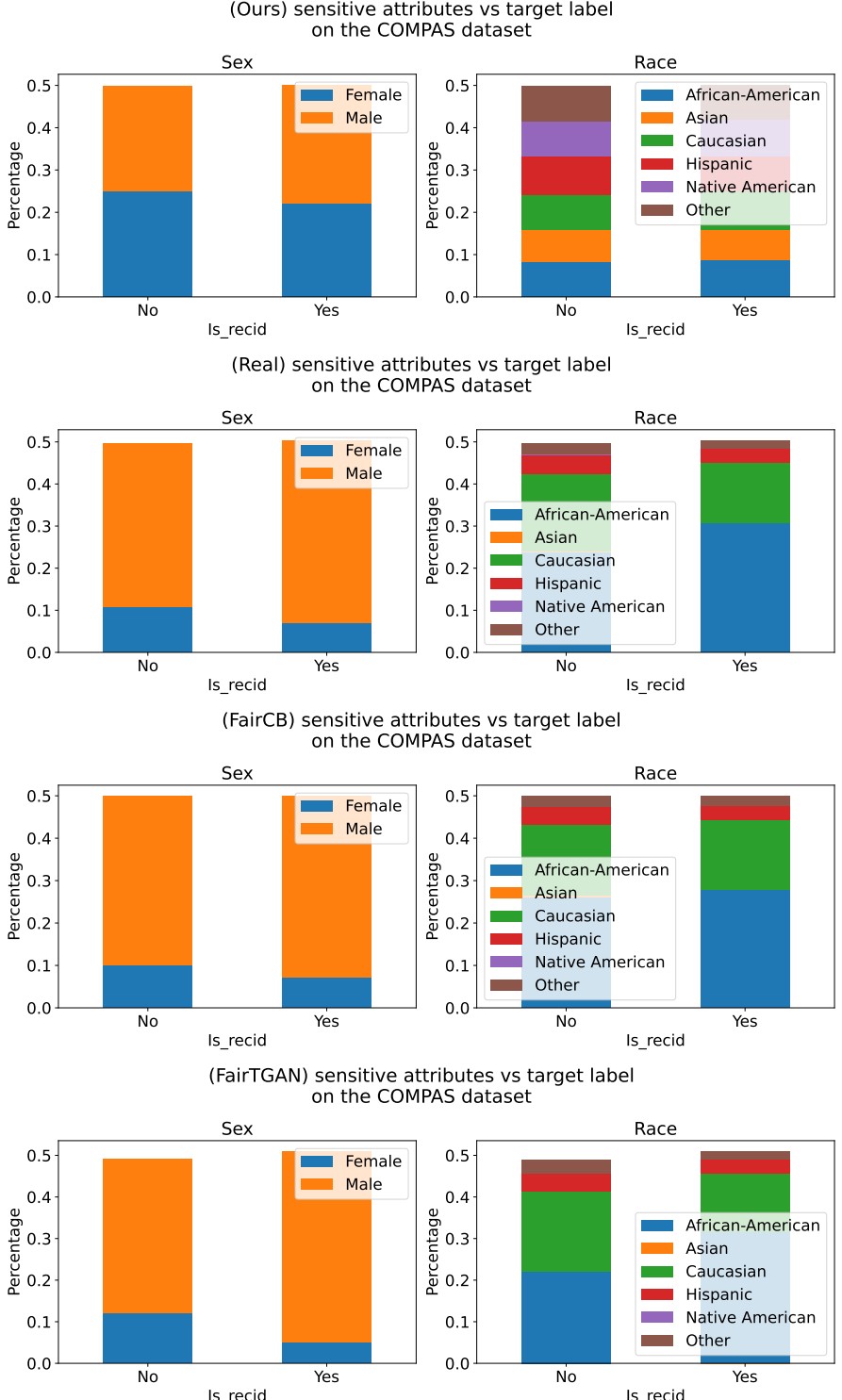

Figure 7: Comparison of models on the COMPAS dataset.

# E  Additional Numerical Results

## E.1  Performance and Fairness Metrics with Sensitive Features Replaced by Uniform Distribution

We present performance and fairness of CatBoost trained on datasets with sensitive attribute replaced by uniform distribution in Table 8 and Table 9 respectively.

Table 8: The values of machine learning efficiency with the best classifier for each dataset. The sensitive attributes are replaced by uniform distribution.

|  | Adult | Bank | COMPAS | Mean |
|---|---|---|---|---|
|  | **AUC (↑)** | | | % |
| Real | $.928_{\pm.000}$ | $.936_{\pm.000}$ | $.799_{\pm.000}$ | 88.8% |
| CoDi | $.858_{\pm.002}$ | $.827_{\pm.015}$ | $.678_{\pm.008}$ | 78.8% |
| GReaT | $.900_{\pm.003}$ | $.690_{\pm.026}$ | $.714_{\pm.010}$ | 76.8% |
| SMOTE | $.914_{\pm.000}$ | $.928_{\pm.001}$ | $.772_{\pm.002}$ | 87.1% |
| STaSy | $.883_{\pm.000}$ | $.894_{\pm.003}$ | $.722_{\pm.014}$ | 83.3% |
| TabDDPM | $.906_{\pm.001}$ | $.917_{\pm.002}$ | $.745_{\pm.002}$ | 85.6% |
| TabSyn | $.910_{\pm.000}$ | $.918_{\pm.001}$ | $.745_{\pm.001}$ | 85.8% |
| FairCB | $.914_{\pm.000}$ | $.907_{\pm.001}$ | $.765_{\pm.002}$ | 86.2% |
| FairTGAN | $.885_{\pm.002}$ | $.873_{\pm.006}$ | $.703_{\pm.002}$ | 82.0% |
| Ours | $.895_{\pm.001}$ | $.913_{\pm.002}$ | $.732_{\pm.001}$ | 84.7% |

Table 9: The values of DPR and EOR with best classifier for each dataset. The sensitive attributes are replaced by uniform distribution.

|  | Adult | Bank | COMPAS | Mean | Adult | Bank | COMPAS | Mean |
|---|---|---|---|---|---|---|---|---|
|  | **DPR (↑)** | | | % | **EOR (↑)** | | | % |
| Real | $.307_{\pm.004}$ | $.415_{\pm.008}$ | $.790_{\pm.005}$ | 50.4% | $.192_{\pm.008}$ | $.368_{\pm.008}$ | $.857_{\pm.007}$ | 47.2% |
| CoDi | $.345_{\pm.030}$ | $.402_{\pm.063}$ | $.901_{\pm.011}$ | 54.9% | $.328_{\pm.038}$ | $.556_{\pm.125}$ | $.923_{\pm.010}$ | 60.2% |
| GReaT | $.256_{\pm.010}$ | $.622_{\pm.269}$ | $.814_{\pm.024}$ | 56.4% | $.169_{\pm.017}$ | $.405_{\pm.079}$ | $.802_{\pm.055}$ | 45.9% |
| SMOTE | $.331_{\pm.017}$ | $.413_{\pm.025}$ | $.777_{\pm.028}$ | 50.7% | $.276_{\pm.045}$ | $.359_{\pm.026}$ | $.826_{\pm.041}$ | 48.7% |
| STaSy | $.304_{\pm.072}$ | $.532_{\pm.060}$ | $.780_{\pm.072}$ | 53.9% | $.240_{\pm.106}$ | $.542_{\pm.065}$ | $.739_{\pm.073}$ | 50.7% |
| TabDDPM | $.281_{\pm.004}$ | $.341_{\pm.017}$ | $.750_{\pm.031}$ | 45.7% | $.191_{\pm.010}$ | $.339_{\pm.010}$ | $.816_{\pm.040}$ | 44.9% |
| TabSyn | $.286_{\pm.005}$ | $.319_{\pm.018}$ | $.815_{\pm.016}$ | 47.3% | $.182_{\pm.011}$ | $.285_{\pm.022}$ | $.847_{\pm.016}$ | 43.8% |
| FairCB | $.305_{\pm.005}$ | $.623_{\pm.011}$ | $.762_{\pm.023}$ | 56.3% | $.227_{\pm.006}$ | $.680_{\pm.021}$ | $.824_{\pm.023}$ | 57.7% |
| FairTGAN | $.481_{\pm.021}$ | $.812_{\pm.115}$ | $.692_{\pm.026}$ | 66.2% | $.544_{\pm.036}$ | $.567_{\pm.169}$ | $.732_{\pm.024}$ | 61.4% |
| Ours | $.503_{\pm.017}$ | $.690_{\pm.085}$ | $.781_{\pm.006}$ | 65.8% | $.580_{\pm.027}$ | $.631_{\pm.087}$ | $.781_{\pm.015}$ | 66.4% |

