# OpenReview forum: "Balanced Mixed-Type Tabular Data Synthesis with Diffusion Models"
_TMLR — Accepted by TMLR_

### Review · Reviewer_maQ4 · 2024-11-28

**Summary Of Contributions:**

1. The paper proposes a novel diffusion-based framework tailored for mixed-type tabular data synthesis, incorporating fairness as a core design principle.
2. It introduces sensitive guidance mechanisms to balance joint distributions of sensitive attributes (e.g., sex, race) and the target label.
3. Demonstrates superior performance in fairness metrics, such as demographic parity ratio (DPR) and equalized odds ratio (EOR), with improvements exceeding 10% compared to state-of-the-art models.
4. Extends the capabilities of diffusion models to accommodate multiple sensitive attributes, enhancing their applicability in fairness-sensitive domains like healthcare and finance.

**Audience:**

Yes

**Broader Impact Concerns:**

No immediate ethical concerns are anticipated from this research.

**Claims And Evidence:**

Yes

**Requested Changes:**

1. Clarify the Contribution Compared to CoDi

While the paper extends conditions to features and generates fair data from a sensitive attributes perspective, the handling of categorical and continuous variables separately does not appear to be a significant novelty since this approach was previously introduced in CoDi. The authors should explicitly clarify how their approach advances beyond CoDi in terms of novelty and effectiveness.

I suggest revising Section 1 and Related Work to clearly delineate the contribution, emphasizing the extension from label-only conditioning to multivariate feature-level conditioning for fairness.

2. Mathematical Justification for Multivariate Guidance

Equation (6) introduces multivariate guidance for conditions, extending the noise estimation to include multiple sensitive attributes. However, the mathematical justification or derivation of this formulation is unclear. Providing a theoretical foundation or citing relevant work would strengthen the validity of this approach.

3. Add Mathematical Recap of Classifier-Free Guidance in Sections 2 or 3

Classifier-free guidance is a fundamental aspect of the proposed method but is only briefly mentioned in the context of multivariate extensions. A concise recap of its mathematical formulation would improve accessibility and clarity, especially for readers unfamiliar with the concept.

Including a brief mathematical explanation of classifier-free guidance in Section 2 or 3 could be beneficial. For instance, highlight how classifier-free guidance is applied and extended to multivariate conditions in Equation (6).

**Strengths And Weaknesses:**

The paper extends the conditioning mechanism from labels to features, enabling fair synthetic data generation by accounting for sensitive attributes. It achieves significant improvements in fairness metrics, such as demographic parity ratio, while maintaining comparable efficacy to state-of-the-art models. This balance between fairness and performance underscores the practical value of the proposed approach. Overall, the work addresses a critical gap in fairness-aware data synthesis.

The paper lacks sufficient mathematical justification for key components, such as the multivariate extension in Equation (6), which weakens the theoretical grounding of the proposed approach. Additionally, the distinction from prior work, particularly CoDi, is not clearly articulated, making the contribution less pronounced. A minor issue is the absence of a clear definition for $\ominus$ in the formulation of $\mu(\mathbf{c}, \mathbf{s}; w_s, \lambda)$, which could hinder reproducibility and clarity.

---

> ### Author Response · Authors · 2025-01-03
> **Official Comment by Authors**
>
> We sincerely thank the reviewer for thoughtfully evaluating our work and providing insightful feedback.
>
> **NOTE: Changes made based on your feedback are highlighted in *GREEN* text.**
>
> **[W1]** The paper lacks sufficient mathematical justification for key components, such as the multivariate extension in Equation (6), which weakens the theoretical grounding of the proposed approach.
>
> - We have added mathematical justification for Equation (6) in Section 4.1.
>
> **[W2]** Additionally, the distinction from prior work, particularly CoDi, is not clearly articulated, making the contribution less pronounced.
>
> - Thanks for your suggestion. We have added the distinction from CoDi in Section 1 to highlight that we extend label-only conditioning to multivariate feature-level conditioning for fairness.
>
> **[W3]** A minor issue is the absence of a clear definition for $\ominus$ in the formulation of $\mu\left(\mathbf{c}, \mathbf{s} ; w_{s}, \lambda\right)$, which could hinder reproducibility and clarity.
>
> - We have added that it means element-wise subtraction in Section 4.1.
>
> **[C1]** Clarify the Contribution Compared to CoDi. While the paper extends conditions to features and generates fair data from a sensitive attributes perspective, the handling of categorical and continuous variables separately does not appear to be a significant novelty since this approach was previously introduced in CoDi. The authors should explicitly clarify how their approach advances beyond CoDi in terms of novelty and effectiveness. I suggest revising Section 1 and Related Work to clearly delineate the contribution, emphasizing the extension from label-only conditioning to multivariate feature-level conditioning for fairness.
>
> - We have revised Section 1 to clearly explain the contribution and in particular the difference between our method and CoDi.
> - We also emphasized the extension from label-only conditioning to multivariate feature-level conditioning for fairness in Section 1.
>
> **[C2]** Mathematical Justification for Multivariate Guidance. Equation (6) introduces multivariate guidance for conditions, extending the noise estimation to include multiple sensitive attributes. However, the mathematical justification or derivation of this formulation is unclear. Providing a theoretical foundation or citing relevant work would strengthen the validity of this approach.
>
> - We have added mathematical justification for multivariate guidance in Section 4.1.
>
> **[C3]** Add Mathematical Recap of Classifier-Free Guidance in Sections 2 or 3. Classifier-free guidance is a fundamental aspect of the proposed method but is only briefly mentioned in the context of multivariate extensions. A concise recap of its mathematical formulation would improve accessibility and clarity, especially for readers unfamiliar with the concept. Including a brief mathematical explanation of classifier-free guidance in Section 2 or 3 could be beneficial. For instance, highlight how classifier-free guidance is applied and extended to multivariate conditions in Equation (6).
>
> - We have added mathematical recap of classifier-free guidance and its extension to multivariate guidance in Sections 3.4.

---

### Review · Reviewer_VFdu · 2024-12-12

**Summary Of Contributions:**

The paper proposes a method for synthesizing fair tabular data. The proposed method involves a conditioning generation process to construct group-balanced data likely to exhibit better fairness properties on downstream tasks. Experimental results show that the methods can improve the fairness of the generated data while preserving the fidelity to the original data distribution.

**Audience:**

Yes

**Claims And Evidence:**

Yes

**Requested Changes:**

Please see the weaknesses above.

**Strengths And Weaknesses:**

**Strengths**:

1. The paper addresses the important problem of debiasing synthetic generation.
2. The document is well-written and relatively easy to follow.
3. The proposed method is sound and intuitive.
4. The experiments are well conducted and demonstrate the efficiency of the proposed method in terms of fairness and fidelity to the real distribution compared to existing methods.



**Weakness**:

1. The related work section can be improved by thoroughly discussing other fair synthetic data generation methods beyond diffusion models

2. While the proposed method is more computationally expensive (i.e., the use of attention mechanism), the gain in fairness and fidelity is not consistent across datasets and fairness metrics.  Compared to some existing methods, the benefit of the proposed method is not thoroughly provided. In particular, methods such as GReaT can be conditioned at test time to generate group-balanced datasets. Under this setup, the significance of the proposed method for fairness is unclear.

3. The method does not provide a means to control the tradeoff between fairness and accuracy. This feature can be handy to practitioners interested in improving fairness while preserving a certain level of business needs.

4. It remains unclear how the method ensures balanced sampling at test time. Does the conditional sampling occur at test time to ensure a group-balanced synthetic dataset, or does every random sampling at test time yield a balanced dataset without further conditioning?

5. The proposed method heavily relies on specified group labels for conditioning, making it less applicable in broader setups where demographic information is unknown due to privacy restrictions.

---

> ### Author Response · Authors · 2025-01-03
> **Official Comment by Authors**
>
> We sincerely thank the reviewer for thoughtfully evaluating our work and providing insightful feedback.
>
> **NOTE: Changes made based on your feedback are highlighted in *PURPLE* text.**
>
> **[W1]** The related work section can be improved by thoroughly discussing other fair synthetic data generation methods beyond diffusion models.
>
> - Section 2.2 has been expanded to include a comprehensive overview of fair synthetic data generation approaches beyond diffusion models.
> - We have organized and introduced methods using GANs, SMOTE, and knowledge distillation in Section 2.2.
>
> **[W2]** While the proposed method is more computationally expensive (i.e., the use of attention mechanism), the gain in fairness and fidelity is not consistent across datasets and fairness metrics. Compared to some existing methods, the benefit of the proposed method is not thoroughly provided. In particular, methods such as GReaT can be conditioned at test time to generate group-balanced datasets. Under this setup, the significance of the proposed method for fairness is unclear.
>
> - From Table 3, our method achieves an average AUC of 84.7%, which is higher than FairTGAN and only 1.6% lower than FairCB. Although altering the original sensitive distribution is expected to reduce AUC, the results demonstrate that our method effectively preserves high fidelity in the generated synthetic data. We have added some explanations in Section 5.4.1.
> - As shown in Table 4, our method significantly surpasses the strongest baselines in fairness metrics. It achieves an average DPR of 68.4% and EOR of 71.4%, both exceeding the next-best results by at least 10%. By jointly optimizing for performance and fairness, our method presents a more balanced tradeoff than existing approaches. We have added some explanations in Section 5.4.2.
> - GReaT’s autoregressive model can conditionally generate group-balanced datasets at test time. However, its AUC (76.9%) is notably lower than ours. Diffusion models are generally more effective for structured data synthesis compared to autoregressive models such as GReaT. We have discussed the use of GReaT for fair synthetic data generation in Section 2.3.
> - Additionally, our architecture supports scalability to multimodal data synthesis (e.g., generating fair images and tabular data), offering a broader range of applications than GReaT. We have addressed this point in the end of Section 1.
>
> **[W3]** The method does not provide a means to control the tradeoff between fairness and accuracy. This feature can be handy to practitioners interested in improving fairness while preserving a certain level of business needs.
>
> - We have introduced a balancing level to measure the degree of transformation applied to the original joint distribution of the target and sensitive attributes, aiming to approximate a uniform joint distribution. Experiments were conducted across different balancing levels.
> - Analyses of performance-fairness trade-off have been included in Section 5.4.4.
>
> **[W4]** It remains unclear how the method ensures balanced sampling at test time. Does the conditional sampling occur at test time to ensure a group-balanced synthetic dataset, or does every random sampling at test time yield a balanced dataset without further conditioning?
>
> - The conditional sampling occurs at test time to ensure a group-balanced synthetic dataset. We have add explanations in Section 4.3.
>
> **[W5]** The proposed method heavily relies on specified group labels for conditioning, making it less applicable in broader setups where demographic information is unknown due to privacy restrictions.
>
> - We have added this limitation and indicate possible future directions such as synthetic attribute imputation in Section 6.

---

### Review · Reviewer_oHWL · 2024-12-22

**Summary Of Contributions:**

This paper studies the problem of mixed-type tabular synthetic data generation. The authors propose a guidance mechanism inspired by classifier free guidance to account for sensitive attributes during the generation time. The authors focus on synthesizing balanced data according to the sensitive attributes. Authors study the proposed method by comparing to several baselines and show that their approach improves fairness metrics.

**Audience:**

Yes

**Broader Impact Concerns:**

The authors provide a broader impact statement.

**Claims And Evidence:**

Yes

**Requested Changes:**

- Please define µ(c,s; ws,λ) in equation 6.
- Please make appropriate citations for the Bank Marketing and COMPAS datasets.
- Can you provide more explanation for Figure 2 in the paper?
- I believe we need additional explanations for the hyper-paramter tuning.
- Optional: a new experiment that studies the effectiveness of your method in the presence of multiple sensitive attributes.

**Strengths And Weaknesses:**

### Strengths:
- The paper reads very well. It was easy to follow the background section and the overall flow of the paper.
- The paper studies an important problem which has an important application in many fields including the healthcare.
- The method has shown to improve the fairness metrics in 3 datasets.
- Comprehensive comparison against other methods.
- Overall the paper is well-written with promising results.

### Weaknesses and questions:
- The proposed framework is claimed to be general and applicable to the case where there are many sensitive attributes, however the authors only experimented with 2 attribute at a time.
- The method assumes that the target in the input data is already balanced. How do we expect this method work in the contexts where the data is not balanced?
- Although the authors report the values of gamma and lambda, it is not clear how those values are selected. Are they easy to tune?
- For the case where we have many sensitive attributes, how can we tune all these hyper-parameters for every attribute without sacrificing the quality of the generations?
- The hyper-parameters are reported in the appendix, are they the same across all the datasets?
- In general, since synthetic data generation is an expensive task and requires us to generate the data given a set of hyper parameters then validate the results, HP tuning is particularly expensive. How does your hyper-paramter tuning compare to other methods?

---

> ### Author Response · Authors · 2025-01-03
> **Official Comment by Authors**
>
> We sincerely thank the reviewer for thoughtfully evaluating our work and providing insightful feedback.
>
> **NOTE: Changes made based on your feedback are highlighted in *BLUE* text.**
>
> **[W1]** The proposed framework is claimed to be general and applicable to the case where there are many sensitive attributes, however the authors only experimented with 2 attribute at a time.
>
> - We acknowledge the absence of experiments involving more than two sensitive attributes and have documented this as a limitation in Section 6.
>
> **[W2]** The method assumes that the target in the input data is already balanced. How do we expect this method work in the contexts where the data is not balanced?
>
> - The experimental datasets used in this study have balanced target distributions.
> - Our method can also generate balanced target distributions and is expected to effectively mitigate bias arising from target imbalance.
>
> **[W3]** Although the authors report the values of gamma and lambda, it is not clear how those values are selected. Are they easy to tune?
>
> - $\gamma$ is an intermediate variable not a hyper-parameter.
> - $\lambda = 1$ as reported in Appendix B and it is selected based on heuristics.
> - These hyper-parameters can be tuned using Optuna, although the process is time-intensive. We employed the hyper-parameters reported in Appendix B, which yielded superior results compared to baseline methods, and therefore did not further fine-tune each parameter individually.
>
> **[W4]** For the case where we have many sensitive attributes, how can we tune all these hyper-parameters for every attribute without sacrificing the quality of the generations?
>
> - To address cases involving multiple sensitive attributes, hyper-parameters can be optimized using a weighted average of fairness and performance scores as the objective function.
> - We have added some content about performance-fairness trade-off in Section 5.4.4.
>
> **[W5]** The hyper-parameters are reported in the appendix, are they the same across all the datasets?
>
> - For DL-based methods, we used the original hyper-parameter settings provided in the original implementations and mainly fine-tuned the number of epochs and learning rate.
> - For SMOTE-based methods, we fine-tuned the number of nearest neighbors.
> - We have added search space of hyper-parameter for baseline methods in Appendix B.
>
> **[W6]** In general, since synthetic data generation is an expensive task and requires us to generate the data given a set of hyper parameters then validate the results, HP tuning is particularly expensive. How does your hyper-parameter tuning compare to other methods?
>
> - Our method has a moderate number of parameters (482,545). This is more than TabDDPM but less than most deep learning models, except FairTGAN.
> - The training based on our method requires 11.8 seconds per epoch, which is similar to CoDi, but slower than TabSyn.
> - However, our method has a long sampling time of 1,213.9 seconds per dataset. We have indicated that improving efficiency could be a future direction in Section 6.
>
> **[C1]** Please define $\mu(c, s; w_s, \lambda)$ in equation 6.
>
> - We have added explanations for $\mu(c, s; w_s, \lambda)$ in Section 4.1.
>
> **[C2]** Please make appropriate citations for the Bank Marketing and COMPAS datasets.
>
> - We have made the Bank Marketing and COMPAS datasets cited in Appendix A.2 and Appendix A.3.
>
> **[C3]** Can you provide more explanation for Figure 2 in the paper?
>
> - Figure 2 illustrates that the synthetic data generated by our approach demonstrates a balanced joint distribution between sensitive features and the target label, effectively reducing the disparities observed in the original data.
> - We have added the explanations for Figure 2 in Section 5.4.3.
>
> **[C4]** I believe we need additional explanations for the hyper-parameter tuning.
>
> - We have added more details in hyper-parameter tuning in Appendix B and Appendix C.
>
> **[C5]** Optional: a new experiment that studies the effectiveness of your method in the presence of multiple sensitive attributes.
>
> - Thank you for the suggestion. We agree that evaluating the effectiveness of our method with multiple sensitive attributes would offer valuable insights.
> - However, due to the limited availability of large tabular datasets with more than two sensitive attributes and time constraints, we have not conducted such experiments. We plan to explore this further in future work.

---

> > ### Comment · Reviewer_oHWL · 2025-01-06
> >
> > Thank you for answering my questions. I am currently mostly concerned with the sampling time of your method. It takes roughly 20 minutes to generate a single sample which is a major drawback specially when you have additional hyper-parameters to tune.
> > I'm wondering about where the bottleneck comes from? Does changing the GPU device improves the sampling time? Is there anyway to improve the sampling time?
> >
> > I'm happy with the rest of the changes. Although I think experimenting with additional sensitive attributes would have been very useful, I'm leaning more towards acceptance.

---

> > > ### Author Response · Authors · 2025-02-05
> > > **Official Comment by Authors**
> > >
> > > We sincerely thank the reviewer for the thoughtful response. We agree that sampling time is a current limitation of our implementation. While using a more powerful GPU can help, we believe the principal bottlenecks are (1) the long diffusion timesteps needed for high-quality generation and (2) the quadratic complexity of the attention modules in our backbone network.
> > >
> > > Possible strategies to mitigate sampling time include:
> > >
> > > (1) Reducing the number of diffusion timesteps using acceleration techniques such as [Flow-DPM-Solver](https://arxiv.org/abs/2410.10629), which optimizes the reverse process for faster sampling.
> > >
> > > (2) Streamlining the backbone network, for instance by using [linear attention](https://arxiv.org/abs/2006.16236) or employing a pre-trained autoencoder to represent tabular data in a lower-dimensional latent space, thereby curbing computational costs.
> > >
> > > We have incorporated these ideas with references into the second paragraph of Section 6 Limitations and Discussions.

---

### Decision · Action_Editor_i7G8 · 2025-02-03

**Recommendation:** Accept with minor revision

**Comment:**

While the authors have revised the manuscript to respond to the reviewers' comments, I believe that Reviewer oHWL's comments could be addressed more directly in the camera-ready version, as follows:
- More than two sensitive attributes: I think that Section 6 should also acknowledge that more sensitive attributes mean more hyperparameters as the reviewer points out, and that it may be computationally expensive to tune all of these, especially given the method's sampling time.
- Hyperparameter selection: I did not see a direct response to the reviewer's question of whether the values were the same for all datasets.

**Audience:**

The reviewers agree that improving fairness of synthetic tabular data is important, particularly for domains such as healthcare.

**Claims And Evidence:**

This submission proposes a diffusion model with guidance from sensitive attributes to generate tabular data with balanced joint distributions of target labels and sensitive attributes, motivated by fairness. The author rebuttal clarified the distinction from prior work, especially CoDi (Lee at al., 2023), and also clarified that the most significant benefit of the proposed method lies in improved downstream fairness metrics (demographic parity ratio, equalized odds ratio). After the rebuttal, all reviewers were in favor of acceptance, and I agree that the support for the proposal is solid.

---

> ### Author Response · Authors · 2025-02-14
> **Official Comment by Authors**
>
> We appreciate the constructive feedback and the positive evaluation of our work. Below, we address the suggested revisions for the camera-ready version:
>
> 1. **More than two sensitive attributes:** We have acknowledged the computational cost of fine-tuning for more sensitive attributes in the second paragraph of Section 6 Limitations and Discussion.
> 2. **Hyperparameter selection:** The hyperparameter search spaces are the same for all datasets. We have clarified this in Appendix B Reproducibility.
>
> We appreciate the feedback from reviewers, which has helped improve the clarity and completeness of our work.